# PrimeStore MTM and OMNIgene Sputum for the Preservation of Sputum for Xpert MTB/RIF Testing in Nigeria

**DOI:** 10.3390/jcm8122146

**Published:** 2019-12-04

**Authors:** John S. Bimba, Lovett Lawson, Konstantina Kontogianni, Thomas Edwards, Bassey Emanna Ekpenyong, James Dodd, Emily R. Adams, Derek J. Sloan, Jacob Creswell, Jose Dominguez, Luis E. Cuevas

**Affiliations:** 1Zankli Research Centre and Department of Community Medicine, Bingham University, Karu 961105, Nigeria; bimbajs@yahoo.com (J.S.B.);; 2Zankli Medical Center, Abuja 961105, Nigeria; 3Centre for Drugs and Diagnostics, Liverpool School of Tropical Medicine, Liverpool L3 5QA, UKThomas.Edwards@lstmed.ac.uk (T.E.); Emily.Adams@lstmed.ac.uk (E.R.A.); 4Department of Clinical Sciences, Liverpool School of Tropical Medicine, Liverpool L3 5QA, UK; 5School of Medicine, University of St Andrews, St Andrews KY16 9TF, UK; djs26@st-andrews.ac.uk; 6Stop TB Partnership, TB REACH, 1218 Geneva, Switzerland; 7Institut d’Investigació Germans Trias i Pujol, Universitat Autònoma de Barcelona, CIBER Enfermedades Respiratorias (CIBERES), 08916 Badalona, Spain; jadomb@gmail.com

**Keywords:** tuberculosis, diagnosis, OMNIGene-sputum, PrimeStore, Xpert MTB/RIF

## Abstract

Background: Xpert MTB/RIF (GX) for tuberculosis (TB) diagnosis is often located in reference laboratories, and sputum needs to be transported using a cold chain. Transport media to preserve sputum are available, but performance data under programmatic conditions are limited. Methods: Sputum samples were collected from patients with presumptive TB in Nigeria. One sputum was transported in a cold chain, tested immediately with GX and cultured. One sputum was swabbed and stored in PrimeStore-Molecular-Transport-Medium (Primestore), and the remainder was stored in OMNIGene-sputum (Omnigene), kept for seven days and tested with GX. Results: Of 248 patients, 63 were fresh-sputum culture-positive and 56 GX-positive (sensitivity 88.9%, 95% CI: 78.4–95.4%). Four of 185 culture-negative patients were GX-positive (specificity 97.8%, 94.6–99.4%). Omnigene GX and Primestore GX were positive in 56/62 (90.3%, 80.1–96.4%) and 49/62 (79.0%, 66.8–88.3%) culture-positive, respectively, and 1/185 (99.5%, 97.0–100.0%) and 3/185 (98.4%, 95.3–99.7%) were culture-negative patients. 14 Human Immunodeficiency Virus (HIV)-infected and 44 HIV-uninfected patients were culture-positive. Omnigene and Primestore detected 12/14 (85.7%, 57.2–98.2%) and 5/14 (35.7%, 12.8–64.9%) HIV-infected and 41/44 (93.2%, 81.3–98.6%) HIV-uninfected culture-positive patients. Interpretation: Omnigene stored and fresh sputum samples had similar GX results. The GX results of Primestore-stored samples were similar to those found in the fresh sputum of non-HIV infected patients, but GX-positivity was lower in HIV-infected patients. This was likely due to the lower amount of bacilli collected by the swab and transferred to PrimeStore.

## 1. Introduction

Tuberculosis (TB) is the leading cause of death due to a single infectious agent globally [1]. National TB Programme (NTP) strategies address this major public health threat by increasing the detection of symptomatic individuals and initiating early treatment. The World Health Organization’s (WHO) End TB strategy and the Stop TB Partnership’s Global Plan to End TB have set ambitious targets for the response to TB, including a 90% reduction in TB incidence by 2035 [2,3]. Meeting these targets will require identifying and testing many more individuals and expanding the reach of tests to health centres without diagnostic facilities. WHO recommends that countries aim to test all people with presumptive TB with Xpert MTB/RIF (GX), to improve the detection and accuracy of TB diagnosis including drug resistance. GX is a molecular test that is usually available in reference or district laboratories [4,5], due to its power and infrastructure needs and machine costs. Expanding access to this test frequently requires the referral of patients or the transport of sputum to testing centres. Sample transportation requires refrigeration to maintain the biological integrity of the sample and prevent bacterial and fungal proliferation, resulting in a decreased culture positivity. Furthermore, although archived clinical samples kept frozen can be later tested with GX with minimal or no loss of sensitivity, data on the performance of GX in samples that have been transported at ambient temperature without preserving reagents are not available in the public domain.

In Nigeria, GX is mostly stationed in district hospitals, Human Immunodeficiency Virus (HIV) clinics, selected health centres and mobile vans. Despite the existing efforts to establish sputum transportation network (such as the ‘Riders for Health’, operating in eight states), these networks are still insufficient to cover the demand for health services. Although data on the time required to transport samples are poorly documented, a review of samples received in our laboratory, Zankli Research Centre, indicated that samples referred from the Northern Region of the country are processed within a median of six days, and 25% are processed within nine days of sample collection.

Implementing a cold chain incurs substantial costs and can be especially challenging in rural areas. The development of sputum transport reagents that eliminate the need for a cold chain during transit could reduce the cost and complexity of sample transportation. One of these reagents, OMNIGene-sputum (Omnigene, DNA Genotek Inc., Canada), is allegedly able to decontaminate and liquefy sputum, whilst preventing sample degradation for up to eight days at ambient temperature [6]. A second reagent, PrimeStore Molecular Transport Medium (Primestore, Longhorn Vaccines & Diagnostics, US) has been reported to inactivate *Mycobacterium tuberculosis* (MTB) and preserve nucleic acids at ambient temperature, for downstream testing [7]. The inactivation of MTB would improve the biosafety of the laboratory and the transport and handling of samples by staff members. Both solutions have been reported to be compatible with molecular tests, including GX [8,9]. Although studies have reported good performance characteristics, most reports include small numbers of samples, typically less than 100 participants, and are sponsored by the manufacturers.

We report here an independent head-to-head evaluation of the effectiveness of both Omnigene and Primestore for GX testing conducted in Abuja, Nigeria, under operational conditions.

## 2. Methods

We enrolled consecutive adults with signs and symptoms of presumptive pulmonary TB attending TB diagnostic clinics at district hospitals within Abuja, as well as patients that had been referred to the TB clinics for diagnosis. Patients receiving TB treatment or who had been treated for TB in the previous two years were excluded. Participants were asked to provide two sputum samples on the day of consultation, with the first sample taken on the spot and the second sample approximately 1 h after the first specimen [10]. One sample was transported in a cold box (called fresh sputum), split into two equal samples after vortexing, tested using GX within 24 h and cultured using Lowenstein–Jensen (LJ) solid media in two tubes at the laboratory. LJ was conducted using the modified Petroff’s method and an N-acetyl-L-cysteine-sodium hydroxide (NALC-NaOH) solution as the digesting and decontaminating agent. The other sample was used for testing the samples stored in Primestore and Omnigene solutions. The samples were not labelled as first or second and were not processed in a particular order, to minimise potential differences in bacterial loads. However, we did not assess the bacterial load distribution in the two sputum cups. For Primestore, a swab was swirled five times through the sputum sample, collecting between 0.1 and 0.2 mL of sputum; placed in the Primestore collection tube; and vortexed to mix the liquid with the sputum. For Omnigene, an equal volume of the Omnigene kit solution was added to the remainder of the sputum sample. Both samples were then transferred to Zankli Research Centre, Bingham University, without refrigeration, and stored for seven days in a room without air conditioning and maximum temperatures ranging from 30 to 37 °C, to mimic the circumstances of field storage and transportation. After seven days, the Omnigene-stored samples were transferred from the cup to a conical tube, vortexed in order to become fully liquid and centrifuged at 3800 g for 20 min. The supernatant was gently poured off without disturbing the sediment, which was then resuspended in sterile Phosphate Buffered Saline for Xpert testing. For Primestore, we transferred 0.7 mL of the “blend” of PrimeStore and sputum into 1.4 mL of the Xpert reagent and transferred the total 2.1 mL volume into the Xpert cartridges. Both samples were then tested with GX and compared with the GX results obtained with the fresh sample and against the culture. The culture was considered the reference standard to estimate the sensitivity, specificity (with 95% confidence intervals, 95% CI) and concordance of GX positivity across the three methods. However, as solid cultures are known to have lower sensitivity than liquid cultures, we also used Kappa statistics to describe the level of agreement between the Xpert results obtained with the three approaches. GX semiquantitative grades were used to reflect differences in bacilli DNA concentrations between the fresh and stored specimens. The sample size was estimated assuming a sensitivity of 0.75 and 070 of GX to identify culture-positive individuals with paired fresh and stored sputum samples with a specificity of 0.95 for both, an expected 20% of culture-positive patients and an expected Kappa of 85%. The study would require a sample size of 202 individuals to achieve a power of 80% and a significance of 5%.

### Ethical Approval

The study was approved by the Health Research Ethics Committees of the Liverpool School of Tropical Medicine, UK, and the Federal Capital Territory, Nigeria (protocol numbers 17–014 and FHREC/2017/01/63/09-08-17, respectively). Patients attending the centres were asked to read and confirm they had understood the study information leaflets and underwent consent procedures. Individuals were included if they willingly provided written informed consent to participate. The corresponding author had full access to all the data included in the study and took responsibility for the integrity of the data and the accuracy of the data analysis. The sample size was estimated assuming a sensitivity of 0.75 and 070 of GX to identify culture-positive individuals with paired fresh and stored sputum samples, respectively, with a specificity of 0.95 for both, an expected 20% of culture-positive patients and a Kappa of 85%. The study would require a sample size of 202 individuals to achieve a power of 80% and a significance of 5%.

## 3. Results

A total of 262 patients were enrolled and 248 (95%) had valid culture results. Of these, 63 (25.4%) were culture-positive and 185 culture-negative (Table 1). GX in fresh sputum detected 56 of 63 culture-positive samples (sensitivity 88.9%, 95% CI: 78.4–95.4%). Sixty-two of the 63 culture-positive patients were also tested with GX, using the Omnigene-stored and Primestore-stored samples. Both sets of samples were GX-positive in 56 samples (sensitivity 90.3%, 95% CI: 80.1–96.4%) and 49 samples (sensitivity 79.0%, 95% CI: 66.8–88.3%) respectively. The sensitivity using Primestore was lower when compared with the fresh sample, but this was not statistically significantly (McNemar’s test, p = 0.070). GX in the 185 culture-negative patients was positive in five patients, including four fresh samples (specificity 97.8%, 95% CI: 94.6–99.4%), one Omnigene-stored sample (specificity 99.5%, 95% CI: 97.0–100.0%) and three Primestore-stored samples (98.4%, 95% CI: 95.3–99.7%). One of these culture-negative samples was positive for all GX tests (fresh GX, Omnigene GX and Primestore GX), suggesting that the culture may have failed to identify the bacteria. There were also no statistically significant differences in specificity between the three methods.

HIV status was known for 59 culture-positive patients, with 14 being HIV-infected and 45 HIV-uninfected. GX was positive in 12 fresh sputum samples (sensitivity 85.7%, 95% CI: 57.2–98.2%), 12 Omnigene-stored samples and five Primestore-stored samples (sensitivity 35.7%, 95% CI: 12.8–64.9%) of the 14 HIV-infected culture-positive patients. Among the 45 culture-positive HIV-uninfected patients, GX was positive in 41/45 fresh sputum samples (sensitivity 91.1%, 95% CI: 78.8–97.5%), 41/44 Omnigene-stored samples and 41/44 Primestore-stored samples (sensitivity 93.2%, 95% CI: 81.3–98.6%, respectively).

GX MTB semiquantitative grades obtained using fresh, Omnigene-stored and Primestore-stored samples are shown in Figure 1. Omnigene-stored sputum had higher GX grades than fresh sputum in 14 (21%) samples, similar grades in 34 (51%) samples and lower grades in 19 (28%) samples. In comparison, Primestore-stored samples had higher GX grades than fresh sputum in four (6%) samples, similar grades in 29 (43%) and lower grades in 35 (51%). 

The agreement between GX conducted in fresh and stored samples is shown in Table 2. The overall agreement between fresh and Omnigene GX was 98.4% (Kappa 0.948, 95% CI: 0.903–0.993) and 94.7% between fresh and Primestore GX (Kappa 0.850, 95% CI: 0.774–0.926), indicating a high level of agreement. However, six culture-positive samples had concomitant fresh, Omnigene and Primestore GX negative results, suggesting that the DNA concentration may have been low and was missed by all GX tests. One culture-positive patient was fresh GX-negative but Omnigene and Primestore GX-positive. Among culture-negative samples, one sample was fresh and Omnigene and Primestore GX-positive, suggesting that this infection may have been missed by the culture, and a further three samples were culture-negative and fresh GX-positive but Omnigene and Primestore GX-negative, which is difficult to interpret.

## 4. Discussion

Our study adds important new information to a sparse evidence base on the role of sample preservation media in TB diagnosis. Sputum storage using Omnigene for one week in the absence of a cold chain had little impact on the sensitivity and specificity of GX. However, storing sputum with Primestore under the same conditions led to a lower, but not statistically significant, GX positivity among samples from HIV-infected individuals, who are more likely to have paucibacillary TB. 

Effective and resilient sputum transport systems are needed to increase access to WHO-recommended diagnostics for TB. These systems need to be easy to implement at a low cost and effective in preserving the integrity of the sputum, as samples might not be immediately collected, might travel long distances and might have to wait for testing at the destination laboratory. Methods that do not need a cold chain would be of particular interest to NTPs with limited resources and large rural area populations, where the provision of cold chain transport systems are prohibitively expensive or unavailable. The ability to safely store samples during transport, avoiding the wastage of GX cartridges and reducing the number of false-negative tests due to poor sample quality, could have a major effect in improving case detection [11] and in the detection of drug resistance by facilitating testing with GX to obtain a RIF resistance reading. Although poor transport conditions increase culture contamination and decrease positivity, there is a paucity of data describing whether the same deleterious changes occur with nucleic acids for GX testing, and there are no studies comparing transport products versus no product, which makes it difficult to disaggregate the additional effect of the product itself. 

Omnigene was developed to preserve MTB cells for culture, with a reported viability of eight days [12]. Data on the use of Omnigene for downstream culture are mixed, however, with some studies reporting a reduced positivity when compared with the standard NALC-NaOH [13] and others reporting an equivalent performance [14]. Previous smaller studies have shown Omnigene to be compatible with molecular diagnostics such as GX [9], and our findings confirm that testing Omnigene-preserved samples was as accurate as directly testing fresh sputum samples, with minimal change in the cycle threshold (CT) values obtained. An important consideration for Omnigene, however, is the need to centrifuge to obtain a concentrated pellet, as this would prevent its use at peripheral sites, where centrifuges are often unavailable. A further issue is its cost, which is estimated at USD 2.84 [6] per sample. Because the Omnigene fluid needs to be added when the sputum is collected, this is likely to be the minimum cost, as health centres need to stock larger amounts in preparation for the number of sputum samples likely to be collected. Health economic studies are required to determine the cost-benefit of implementing Omnigene within TB programs. 

Primestore was primarily designed to preserve nucleic acids for molecular testing, rather than preserving cells for culture, and contains reagents to lyse mycobacteria, protecting nucleic acids from nuclease-mediated degradation [7]. Samples preserved in Primestore have been utilised for next generation sequencing [15] to identify drug-resistant mutations. This indicates their applicability to downstream genomics that require a level of nucleic acid integrity. In this study, sputum preserved with this solution resulted in lower GX semiquantitative grades and a higher proportion of false-negative samples in HIV-infected patients. As the reduced sensitivity was observed among HIV-infected individuals, it is likely that the smaller volume of sputum collected through the swab resulted in a lower bacillary load. Moreover, it may also result in a reduced reliability of Primestore when testing specimens with GX in HIV-infected patients, compromising the usefulness of this sputum preservation method in some southern African populations. Longhorn Vaccines and Diagnostics, Primestore’s manufacturer, has recognised this issue and currently recommends testing clinical specimens using a proprietary quantitative polymerase chain reaction (qPCR) assay.

We acknowledge that the study has several limitations. These include the unavoidable comparison of test performances using results obtained from different samples, which may have masked natural variations in bacilli loads among the sputum cups. We also used an LJ solid culture as the reference standard. LJ has lower sensitivity than liquid culture but was used because the latter is more prone to contamination in a study setting. Its use may have resulted in missing patients with microbiologically confirmed TB, and this may well be the case among participants with negative culture and multiple GX-positive results. Furthermore, although the study included a larger number of samples than those of previous studies, it was not powered to identify differences between subgroups with low frequencies. This is specifically pertinent with regard to the subgroup of patients co-infected with HIV, as the number of participants (*n* = 14) was too small to demonstrate statistical differences. 

Furthermore, as Omnigene was developed to preserve MTB cells for culture, with a reported viability of eight days, it would have been desirable to test for culture at the seventh day as well, to confirm its performance for both GX and the culture. Moreover, the study aimed to evaluate the effectiveness of both Omnigene and Primestore for GX and shows that the results are similar to a GX test on a fresh sample. Although we opted for the comparison at day 1 and not at day 7, we failed to compare all tests at equal time intervals, which could have generated further evidence regarding when and where to use these reagents. 

## 5. Conclusions

This study demonstrates that Omnigene has the potential to facilitate a cold-chain-free transport system for the collection of sputum for GX testing. Further studies are needed to identify logistics and operational issues and cost-effectiveness when implemented in areas with low access to diagnosis to inform a potential large-scale implementation. These products should also be studied with specimens that require downstream culture, as transport can affect culture results to a higher degree than GX testing. Specifically, evidence is needed to document Omnigene’s value for testing specimens collected from drug-resistant TB patients, for treatment monitoring and for the diagnosis of paucibacillary smear-negative TB. 

## Figures and Tables

**Figure 1 jcm-08-02146-f001:**
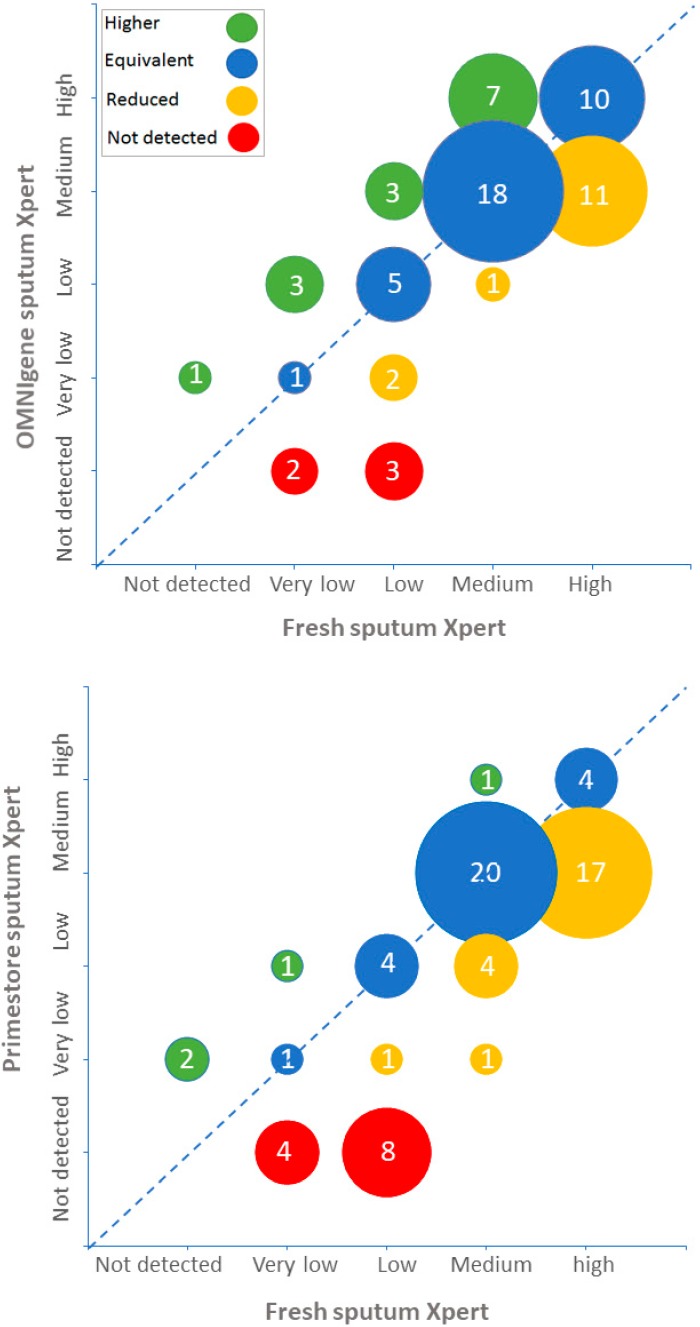
Xpert semiquantitative results of fresh, Omnigene-stored and Primestore-stored sputum specimens.

**Table 1 jcm-08-02146-t001:** Sensitivity and specificity of Xpert MTB/RIF in fresh, Omnigene-stored and Primestore-stored sputum.

Culture		Positive (*n* = 63)		Negative (*n* = 185)		Total	Sensitivity, 95% CI^1^	Specificity, 95% CI
	Xpert MTB/RiF	Positive *n* (%)	Negative *n* (%)	Positive *n* (%)	Negative *n* (%)			
**All**	Fresh	56 (88.9)	7 (11.1)	4 (2.2)	181(97.8)	248	88.9%, 78.4–95.4%	97.8%, 94.6–99.4%
	Omnigene	56 (90.3)	6 (9.7)	1 (0.5)	184(99.5)	247	90.3%, 80.1–96.4%	99.5%, 97–100%
	Primestore	49 (79)	13 (21)	3 (1.6)	181(98.4)	246	79%, 66.8–88.3%	98.4%, 95.3–99.7%
**HIV^1^ infected**	Fresh	12 (85.7)	2 (14.3)	2 (2.8)	70 (97.2)	86	85.7%, 57.2–98.2%	97.2%, 90.3–99.7%
	Omnigene	12 (85.7)	2 (14.3)	1 (1.4)	71 (98.6)	86	85.7%, 57.2–98.2%	98.6%, 92.5–100%
	Primestore	5 (35.7)	9 (64.3)	3 (4.2)	69 (95.8)	86	35.7%, 12.8–64.9%	95.8%, 88.3–99.1%
**HIV negative**	Fresh	41 (85.4)	4 (14.6)	2 (2.0)	94 (98.0)	141	91.1%, 78.8–97.5%	97.9%, 92.7–99.7%
	Omnigene	41 (93.2)	3 (6.8)	0 (0)	96 (100)	140	93.2%, 81.3–98.6%	100.0%, 96.2–100.0%
	Primestore	41 (93.2)	3 (6.8)	0 (0)	96 (100)	140	93.2%, 81.3–98.6%	100.0%, 96.2–100.0%
**HIV Unknown**	Fresh	3 (75)	1 (25)	0 (0)	17 (100)	21	75.0%, 19.4–99.4%	100.0%, 80.5–100.0%
	Omnigene	3 (75)	1 (25)	0 (0)	17 (100)	21	75.0%, 19.4–99.4%	100.0%, 80.5–100.0%
	Primestore	3 (75)	1 (25)	0 (0)	16 (100)	20	75.0%, 19.4–99.4%	100.0%, 79.4–100.0%

^1^ CI = Confidence Interval; HIV =Human Immunodeficiency Virus.

**Table 2 jcm-08-02146-t002:** Agreement of fresh sputum and Omnigene-stored and Primestore-stored Xpert MTB/RIF by culture.

Culture		Positive		Negative		
		Fresh Xpert		Fresh Xpert		
		Positive	Negative	Positive	Negative	Kappa (95% CI)
**Omnigene Xpert**	**positive**	55	1	1	0	0.948 (0.903–0.993)
	**negative**	0	6	3	181	
**Primestore Xpert**	**positive**	48	1	1	2	0.850 (0.774–0.926)
	**negative**	7	6	3	178	

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
