# Peer review of "PrimeStore MTM and OMNIgene Sputum for the Preservation of Sputum for Xpert MTB/RIF Testing in Nigeria"

_jcm, 2019, doi:10.3390/jcm8122146_

Round 1
Reviewer 1 Report
The article focuses on the important challenges represented by the trasport of samples to access diagnostic centers. While the subject is not new, the author has included the comparison of PrimeStore medium, which has been rarely described
Introduction:
The author describes the distribution of Xpert in the laboratory network, however the requirements for sample transport described refer to culture, and not Xpert, which does not mandatorily require cool chain.
Methods:
LJ medium, considered as gold standard, is performed on the same sample used for Xpert reagent-free. Was the sample split? Which decontamination method was used before culture inoculation?
The author compares the performance of Omnigene and PrimeStore to LJ, however these tests are performed on a different sample. Were samples A and B compared to ensure equal bacterial load distribution?
How stratification for HIV status contribute to the evaluation rather than microscipy grade stratification?
Results
line 138: LJ is considered as the gold standard, which contraddicts the conclusion that cultures may have been false negative.
Table 1 is difficult to follow, percentages may be added
Discussion:
The author should expand on study limitations, including comparison of tests performance using results obtained from different samples. This approach may be more justified in a per patient analysis.
Line 219-220: Recent studies have showed performance of Xpert after transport without reagents, the literature review may be updated
Author Response
The author describes the distribution of Xpert in the laboratory network, however the requirements for sample transport described refer to culture, and not Xpert, which does not mandatorily require cool chain.
R: The referee may notice that PrimeStore is designed to maintained DNA integrity and is not designed for culture. This is alluded to in the description of the solutions. We have however edited the text to highlight that there is very little evidence of the effect of poor transport on GX positivity. Edited text reads: “Sample transportation requires refrigeration to maintain the biological integrity of the sample and prevent bacteria and fungal proliferation, resulting in a decreased positivity of culture. Furthermore, although there is very limited data to describe whether similar deleterious changes occur for GX testing, as there are no studies describing its performance in paired samples transported with and without a cold chain, sputum integrity is often require to further test patients with positive GX results that require culture for drug-sensitivity testing.”
Methods:
LJ medium, considered as gold standard, is performed on the same sample used for Xpert reagent-free. Was the sample split? Which decontamination method was used before culture inoculation?
R: Decontamination for culture used the modified Petroff's method, using N-acetyl-L-cysteine-sodium hydroxide (NALC-NaOH) solution as digesting and decontaminating agent. Text amended to say “One sample was transported in a cold box (called fresh sputum), split into two equal samples after vortexing and tested using GX within 24 hours and cultured using Lowenstein Jensen solid media in two tubes at the laboratory. LJ was conducted using the modified Petroff's method and N-acetyl-L-cysteine-sodium hydroxide (NALC-NaOH) solution as the digesting and decontaminating agent.”
The author compares the performance of Omnigene and PrimeStore to LJ, however these tests are performed on a different sample. Were samples A and B compared to ensure equal bacterial load distribution?
R: Samples A and B were not compared to assess bacterial load distribution, although we had stated that “…The samples were not labelled as first or second and were not processed in a particular order.” We have thus expanded this statement to make it more explicit. Edited text reads: “The samples were not labelled as first or second and were not processed in a particular order to minimise potential differences in bacterial loads, although we did not assess the bacterial load distribution in the two sputum cups.”
How stratification for HIV status contribute to the evaluation rather than microscopy grade stratification?
R: This is relative to the context of the reader. In Nigeria, the National TB control Program is using GX as the first test for diagnosis and does not require the use of smear microscopy. HIV-infected patients are often more difficult to diagnose because they have lower bacterial loads and GX has a lower sensitivity. We therefore examined GX results stratified by HIV status and described the semi-quantitative bacterial loads obtained with GX in fresh and preserved samples. We believe this is an appropriate approach for the analysis, which is replicated in other publications.
Results
line 138: LJ is considered as the gold standard, which contradicts the conclusion that cultures may have been false negative.
R: we disagree with this statement. Culture is not a gold standard. It is a reference standard, and as such it has some limitations. For example it is possible to be negative and miss some samples that could have low bacilli loads, or due to selecting a fraction of the sputum sample that did not contain bacilli. If one patient has positive GX in the fresh-, Omnigene- and Primestore- samples, it is possible that the culture had failed to identify the bacteria. We believe our statement is technically correct. We have edited to remove the phrase ‘false-negative’ and indicate: “One of these culture-negative samples was positive for all GX tests (fresh-, Omnigene- and Primestore-GX), suggesting the culture may have failed to identify the bacteria.”
Table 1 is difficult to follow, percentages may be added
R: we have added percentages, as requested. Please notice these paercentages correspond to the sensitivity and specificity described with 95% confidence limits in the same table. The editor may decide to remove them, if considered redundant.
Discussion:
The author should expand on study limitations, including comparison of tests performance using results obtained from different samples. This approach may be more justified in a per patient analysis.
R: we have added a new paragraph of limitations: “We acknowledge that the study has several limitations. These included the unavoidable comparison of tests performance using results obtained from different samples, which may have masked natural variation in bacilli loads among the sputum cups. We also used LJ solid culture as the reference standard. LJ has lower sensitivity than liquid culture, but was used because liquid culture is more prone to contamination in the study setting. Its use may have resulted in missing patients with microbiologically confirmed TB, and this may well be the case among participants with negative culture and multiple GX-positive results. Furthermore, although the study included larger number of samples than previous studies, the study was not powered to identify differences between sub-groups with lower frequencies.”
Line 219-220: Recent studies have showed performance of Xpert after transport without reagents, the literature review may be updated.
R: The referee refers to recent studies reporting the performance of Xpert after transport without reagents. Unfortunately, we were unable to identify these studies in PubMed after many attempts and use of multiple key word combinations. We would be grateful if the referee were able to point to these studies for us, and thus happy to further edit the discussion.
Reviewer 2 Report
Introduction
Line 54. ‘gresh’ should read ‘fresh’
Line 63-76: this is general text, you may mention the situation in Nigeria. For example GX is not only stationed in (district) hospitals but also in health centres, HIV clinics and mobile vans. and there exist a sputum transportation network, e.g. Riders for Health is operating in 8 States. The statement “Sample transportation requires refrigeration to maintain the biological integrity of the sample and prevent bacteria and fungal proliferation”, is in a way true, but very much depends on the time required for the sample transportation and time between collection and examination. And also, what are specific the problems in Nigeria, with regards e.g in getting readable results, what is this proportion currently? How long s the average time between collection and examination? You may elaborate in little more on this in the problem statement
Method
Line 91: “We enrolled 626 adults….”’ This statement is a result, not a method. Pls describe the study population including and the expected numbers to be enrolled.
Line 110-117; “Both samples were then tested with GX, and compared to the GX results obtained with the fresh sample, and against culture…”. Culture was considered gold standard.
Three methods are compared ( fresh sample with an examination within the same day (control), and compared with Omnigene and Primestore stored samples which were tested after 7 days of storing. The comparison could be better, or even could have shown significant differences, if the controls were tested, besides on the first day (fine), also at 7th days while kept under the “field conditions”.
Also, in the light of the statement mention later (line 222) that Omnigene was developed to preserve MTB cells for culture, with reported viability for 8 days, it would have been better to test for culture also at the 7th day. All could even be tested at 2 or 4 days as well. This because in reality, often samples are tested within that range, and not per se after 7 days.
Pls explain what happened to the patients: e.g was the diagnosed established based on the fresh sputum with the GX result on the same day, or did they wait till day 7? Or based on the culture result?
Result
Line 38 : The sentence: “There were no statistically significant differences in specificity among the three methods” suggest that there were significant differences in the sensitivity among the three methods, which is not the case, Therefore I suggest to add the word also “There were ALSO no statistically significant differences in specificity among the three method”.
Discussion 204-246
Line 204 Here starts the discussion, pls insert the title ’discussion’
Line 204-208 the statement “Sputum storage using Omnigene for one week in the absence of a cold chain had little impact on the sensitivity and specificity of GX in this study” is true, but as mentioned above the comparison (control test with in 24 hours versus 7 days) is not a good comparison.
The statement . “However, storing sputum with Primestore under the same conditions led to a decrease in GX positivity among samples from HIV-infected individuals, which are more likely to have paucibacillary TB” is true, but the difference is not significant. Actually there is no any significant difference, all 95% CIs are overlapping.
You may add some lines on the shortcomings of the study. On the method; I already explained, but also on the sample seize: probably ( particularly to see differences among HIV+ and HIV + patients, the sample seize was rather small, and differences in sensitivity could have been significant, f they were larger.
Conclusion
Line 247 Here starts the conclusion, pls insert the title ’conclusion’
It is correct that “ This study demonstrates that Omnigene has the POTENTIAL to facilitate a cold-chain-free transport: , but this is not new. The authors already mention that further studies are needed to inform the potential large scale implementation in areas with low access to diagnosis. Where access is poor the “sample-to laboratory -time” might be large, unfortunately the method used (comparing 7 days to one day) , does not show at which day during the transport-time ( e.g. 2nd day”4th day? ), Omnigene stored sputum would show a significant benefit as compared to storing without.
Final remark
The study aims to evaluation of the effectiveness of both Omnigene and Primestore for GX testing conducted in Abuja, Nigeria, under operational conditions. You have shown that results with O and P are similar to a GX test done at the same day. The article may improve in strength when you explicit mention why you opted for the comparison at one day and not also at day 7. If you didn’t think about that, than indeed mention it under shortcomings of the study design. The method is not wrong, it shows that there is a good effect after 7 days, but I consider it as a missed opportunity not to examine with all three test at same time intervals, this could have informed NTP managers with more evidence when and where to use these reagents.
Author Response
Line 54. ‘gresh’ should read ‘fresh’
edited
Line 63-76: this is general text, you may mention the situation in Nigeria. For example GX is not only stationed in (district) hospitals but also in health centres, HIV clinics and mobile vans. and there exist a sputum transportation network, e.g. Riders for Health is operating in 8 States. The statement “Sample transportation requires refrigeration to maintain the biological integrity of the sample and prevent bacteria and fungal proliferation”, is in a way true, but very much depends on the time required for the sample transportation and time between collection and examination. And also, what are specific the problems in Nigeria, with regards e.g in getting readable results, what is this proportion currently? How long s the average time between collection and examination? You may elaborate in little more on this in the problem statement
R: Thank you for the suggestion. We have edited the text to read: ‘In Nigeria GX is mostly stationed in district hospitals, HIV clinics and selected health centres and mobile vans. Although there are efforts to establish sputum transportation networks, such as the ‘Riders for Health’ operating in eight States, these networks are still insufficient to cover the demand from health services. Although data on the time require to transport samples is poorly documented, a review of samples received in our laboratory, Zankli Research Centre, indicated that samples referred from the Northern Region of the country are processed within a median of 6 days and 25% are processed after 9 days of sample collection (data not published). ‘
Method
Line 91: “We enrolled 626 adults….”’ This statement is a result, not a method. Pls describe the study population including and the expected numbers to be enrolled.
R: We have moved the number to results and text is edited to read: ‘We enrolled consecutive adults attending TB diagnostic clinics at district hospitals within Abuja with signs and symptoms of presumptive pulmonary TB and patients that had been referred to the TB clinics for diagnosis.’
Line 110-117; “Both samples were then tested with GX, and compared to the GX results obtained with the fresh sample, and against culture…”. Culture was considered gold standard. Three methods are compared ( fresh sample with an examination within the same day (control), and compared with Omnigene and Primestore stored samples which were tested after 7 days of storing. The comparison could be better, or even could have shown significant differences, if the controls were tested, besides on the first day (fine), also at 7th days while kept under the “field conditions”.
R: This is correct, and we agree the implication that we could have found agreements and disagreement only by adding the solutions (independently of the time stored). However, we aimed to test whether the results varied after 7 days and we did not test on day 1. In practise this would have been difficult, as we would have had to collect a further sputum sample because stored samples were not opened during storage to avoid the risk of contamination.
Also, in the light of the statement mention later (line 222) that Omnigene was developed to preserve MTB cells for culture, with reported viability for 8 days, it would have been better to test for culture also at the 7th day. All could even be tested at 2 or 4 days as well. This because in reality, often samples are tested within that range, and not per se after 7 days.
R: again, we agree with these statements. Unfortunately, we did not culture on day 7 and testing sputum for culture at 1, 2, 4, 7 days is logistically difficult given the volume os specimens required. We have added this statement to the limitations of the study. We added the discussion statement ‘Furthermore, as Omnigene was developed to preserve MTB cells for culture, with reported viability for 8 days, it would have been desirable to test for culture also at the seventh day to confirm its performance for both GX and culture.’
Pls explain what happened to the patients: e.g was the diagnosed established based on the fresh sputum with the GX result on the same day, or did they wait till day 7? Or based on the culture result?
R: The diagnosis and clinical management was based on the GX result obtained on the fresh sputum. The clinic records of patients with positive culture results but negative GX were reviewed to confirm whether they had initiated treatment and were contacted by phone if necessary. This statement is added to the ethics statement,
Result
Line 38 : The sentence: “There were no statistically significant differences in specificity among the three methods” suggest that there were significant differences in the sensitivity among the three methods, which is not the case, Therefore I suggest to add the word also “There were ALSO no statistically significant differences in specificity among the three method”.
R: amended
Discussion 204-246
Line 204 Here starts the discussion, pls insert the title ’discussion’
R: Title moved from line 203 to make it clearer.
Line 204-208 the statement “Sputum storage using Omnigene for one week in the absence of a cold chain had little impact on the sensitivity and specificity of GX in this study” is true, but as mentioned above the comparison (control test with in 24 hours versus 7 days) is not a good comparison.
R: this statement has been discussed above. Unfortunately the alternative study design suggested cannot be amended at this stage.
The statement . “However, led to a decrease in GX positivity among samples from HIV-infected individuals, which are more likely to have paucibacillary TB” is true, but the difference is not significant. Actually there is no any significant difference, all 95% CIs are overlapping.
R: The referee is correct as the CI overlap. Text edited to say: ‘However, storing sputum with Primestore under the same conditions led to a lower, but not statistically significant, GX positivity among samples from HIV-infected individuals, which are more likely to have paucibacillary TB.
You may add some lines on the shortcomings of the study. On the method; I already explained, but also on the sample seize: probably ( particularly to see differences among HIV+ and HIV + patients, the sample seize was rather small, and differences in sensitivity could have been significant, f they were larger.
R: referee 1 made a similar suggestion and we have added statements on the limitations of the study. Revised paragraph reads: ‘We acknowledge that the study has several limitations. These included the unavoidable comparison of tests performance using results obtained from different samples, which may have masked natural variation in bacilli loads among the sputum cups. We also used LJ solid culture as the reference standard. LJ has lower sensitivity than liquid culture, but was used because liquid culture is more prone to contamination in the study setting. Its use may have resulted in missing patients with microbiologially confirmed TB, and this may well be the case among participants with negative culture and multiple GX-positive results. Furthermore, although the study included larger number of samples than previous studies, the study was not powered to identify differences between sub-groups with lower frequencies. Furthermore, as Omnigene was developed to preserve MTB cells for culture, with reported viability for 8 days, it would have been desirable to test for culture also at the seventh day to confirm its performance for both GX and culture.’
Conclusion
Line 247 Here starts the conclusion, pls insert the title ’conclusion’
R: amended
It is correct that “ This study demonstrates that Omnigene has the POTENTIAL to facilitate a cold-chain-free transport: , but this is not new. The authors already mention that further studies are needed to inform the potential large scale implementation in areas with low access to diagnosis. Where access is poor the “sample-to laboratory -time” might be large, unfortunately the method used (comparing 7 days to one day) , does not show at which day during the transport-time ( e.g. 2nd day”4th day? ), Omnigene stored sputum would show a significant benefit as compared to storing without.
R: this is a similar statement to the comments for lines 204-208. As indicated, this study highlights its potential after 7 days, and does not aim to investigate whether it would be useful at 2-4 days of storage. We would be happy to review the statement if this is still needed. The referee’s statement does not indicate that changes are needed.
Final remark
The study aims to evaluation of the effectiveness of both Omnigene and Primestore for GX testing conducted in Abuja, Nigeria, under operational conditions. You have shown that results with O and P are similar to a GX test done at the same day. The article may improve in strength when you explicit mention why you opted for the comparison at one day and not also at day 7. If you didn’t think about that, than indeed mention it under shortcomings of the study design. The method is not wrong, it shows that there is a good effect after 7 days, but I consider it as a missed opportunity not to examine with all three test at same time intervals, this could have informed NTP managers with more evidence when and where to use these reagents.
R: limitation added to the paragraph on limitations, as suggested.
Round 2
Reviewer 1 Report
Thank you for your edits and comprehensive replies.
I agree with the culture limitations you described, however you use it for comparison and calculation of your sensitivity and specificity, which may be debatable.
The introduction could be clearer, and better decribe the requirements for storage for Xpert.
Author Response
I agree with the culture limitations you described, however you use it for comparison and calculation of your sensitivity and specificity, which may be debatable.
R: Thank you for the comment. As stated, any reference standard has limitation and prior knowledge of its performance is important to interpret results. We agree it is debatable, but a pragmatic approach within the context of Sub-Saharan Africa. The referee may be interested in our publication on this topic (doi: 10.1186/1756-0500-6-215. , Comparison of Mycobacterium tuberculosis drug susceptibility using solid and liquid culture in Nigeria.) in which we describe a substantial degree of agreement between solid and liquid culture, but that using the two methods in tandem increased the number of culture-positive patients as liquid culture also missed patients due to a higher proportion of contaminated specimens.
The introduction could be clearer, and better describe the requirements for storage for Xpert.
R: We have tried to be more explicit. We have tried once again to find more information on the GLI manuals, the Xpert manufacturer and publications to identify recommendations for sputum that will be tested with GX. The referee, we are sure, is familiar with the general recommendations of using a cold chain, to test within seven days etc. However, these recommendations are mostly focused on culture and are vague about GX testing. We tried to find evidence from the literature. However, there are very few papers focusing on this issue. Some publications indicate stored samples kept frozen (at -40 to -80C) seem to maintain DNA and can be tested with GX. With minimal loss of sensitivity after 2.5 years. However, we could not identify any publication that investigates whether GX performance changes if the sputum is not refrigerated (in any direction). We have thus stated the sentence below in the introduction: "Furthermore, although archived clinical samples kept frozen can be later tested with GX with minimal or no loss of sensitivity, data on the performance of GX in samples that have been transported at ambient temperature without preserving reagents is not available in the public domain."
Interesting though and we have set up a prospective study to see if GX positive samples that are not refrigerated continue to be positive after 1, 2, 3 and 5 weeks of collection.
Reviewer 2 Report
The paper has significantly improved. My only comment at this stage is to include in the method section the expected number of patients to be enrolled in the study population.
The reason why storing sputum with Primestore under the same conditions led to a lower, but not statistically significant, decrease in GX positivity among samples from HIV-infected individuals, is probably because the small number of HIV-infected cases. Therefore it is important to show your initial expected number of the study population. If this number is close to the 626, than you may add in the limitations section, that the sample seize was to small to show differences with HIV-infected group (N=14).
Author Response
The paper has significantly improved. My only comment at this stage is to include in the method section the expected number of patients to be enrolled in the study population.
R: We have included a statement for the sample size estimations, as requested. "The sample size was estimated assuming a sensitivity of 0.75 and 070 of GX to identify culture-positive individuals with paired fresh and stored sputum samples, respectively, with specificity of 0.95 for both an expected 20% of patients being culture positive and a Kappa of 85%. The study would require a sample size of 202 individuals to achieve a power of 80% and significance of 5%."
The reason why storing sputum with Primestore under the same conditions led to a lower, but not statistically significant, decrease in GX positivity among samples from HIV-infected individuals, is probably because the small number of HIV-infected cases. Therefore, it is important to show your initial expected number of the study population. If this number is close to the 626, than you may add in the limitations section, that the sample seize was too small to show differences with HIV-infected group (N=14).
R: This limitation was already included in the limitations, as the study is not powered to identify differences among groups. As the referee requested a specific statement, we have edited the text to refer to the HIV group. Revised text reads: “Furthermore, although the study included larger number of samples than previous studies, the study was not powered to identify differences between sub-groups with low frequencies. This is specifically pertinent to the subgroup of patients co-infected with HIV, as the number of participants (N=14) was too small to demonstrate statistical differences.”